# Melatonin Can Modulate Neurodegenerative Diseases by Regulating Endoplasmic Reticulum Stress

**DOI:** 10.3390/ijms24032381

**Published:** 2023-01-25

**Authors:** Yeong-Min Yoo, Seong Soo Joo

**Affiliations:** 1East Coast Life Sciences Institute, College of Life Science, Gangneung-Wonju National University, Gangneung 25457, Republic of Korea; 2Department of Marine Bioscience, College of Life Science, Gangneung-Wonju National University, Gangneung 25457, Republic of Korea

**Keywords:** melatonin, neurons, neurodegenerative diseases, endoplasmic reticulum stress

## Abstract

As people age, their risks of developing degenerative diseases such as cancer, diabetes, Parkinson’s Disease (PD), Alzheimer’s Disease (AD), rheumatoid arthritis, and osteoporosis are generally increasing. Millions of people worldwide suffer from these diseases as they age. In most countries, neurodegenerative diseases are generally recognized as the number one cause afflicting the elderly. Endoplasmic reticulum (ER) stress has been suggested to be associated with some human neurological diseases, such as PD and AD. Melatonin, a neuroendocrine hormone mainly synthesized in the pineal gland, is involved in pleiotropically biological functions, including the control of the circadian rhythm, immune enhancement, and antioxidant, anti-aging, and anti-tumor effects. Although there are many papers on the prevention or suppression of diseases by melatonin, there are very few papers about the effects of melatonin on ER stress in neurons and neurodegenerative diseases. This paper aims to summarize and present the effects of melatonin reported so far, focusing on its effects on neurons and neurodegenerative diseases related to ER stress. Studies have shown that the primary target molecule of ER stress for melatonin is CHOP, and PERK and GRP78/BiP are the secondary target molecules. Therefore, melatonin is crucial in protecting neurons and treating neurodegeneration against ER stress.

## 1. Introduction

The endoplasmic reticulum (ER) is critical for normal cellular function. ER stress due to dysfunction or loss of integrity caused by the accumulation of unfolded proteins or changes in calcium homeostasis within the ER can lead to apoptosis and cell signals associated with apoptosis [1,2,3]. Common causes of neurodegenerative diseases include the accumulation and deposition of misfolded proteins that affect neuronal connectivity, cell death, and faulty cell signaling [4,5]. Unfolded protein response (UPR) activity, typically dysfunctional due to erroneous protein aggregation or oxidative stress in cells, can cause more protein accumulation in the cells, leading to ER stress and disease exacerbation [6,7]. Melatonin, a neuroendocrine hormone mainly synthesized in the pineal gland, is involved in pleiotropically biological functions, including the control of the circadian rhythm, immune enhancement, antioxidant, anti-aging, and anti-tumor effects [8,9].

This paper aimed to collect and analyze data on the association between neurodegenerative diseases and ER stress in PubMed published up to 23 November 2022, focusing on molecular mechanisms involved in such association according to the year of publication and how melatonin might contribute to the relationship between ER stress and human neurological disorders.

## 2. Relationship between Endoplasmic Reticulum Stress and Neurodegenerative Diseases

As of 23 November 2022, the total number of published articles related to neurodegenerative diseases in PubMed was 400,813, of which 17,869 included oxidative stress among the cell mechanisms related to neurodegenerative diseases. There were 13,097 articles on apoptosis, 11,477 on mitochondria, 6481 on autophagy, and 1627 on ER stress. Among the neurodegenerative disease studies, ER stress-related research started in 1990 had fewer studies than the other studies. According to PubMed, there were 514 review articles. In 2005, there were only about 20 papers. However, the research has been quite explosive recently (Figure 1).

According to PubMed, Parkinson’s Disease (PD) was the most studied neurodegenerative disease, with a focus on ER stress, followed by Alzheimer’s Disease (AD), Amyotrophic Lateral Sclerosis (ALS), Transmissible Spongiform Encephalopathy (TSE), polyglutamine diseases, acute neurodegeneration, and neuronal storage diseases (Figure 2). This suggests that PD, AD, and ALS are the most relevant neurodegenerative diseases in the elderly. It also suggests that the research has been focused on ER stress for treating these diseases.

In the following, we will describe how ER stress in neurodegenerative diseases may contribute to the disease process of human neurological disorders.

### 2.1. ER Stress and Parkinson’s Disease (PD)

PD is an age-related, chronic, progressive, degenerative disease caused by the decline of dopaminergic neurons in a specific brain area, i.e., the substantia nigra pars compacta. Typically, PD is characterized by intraneuronal cytoplasmic inclusion bodies, known as Lewy bodies, caused by the abnormal deposition of a protein called alpha-synuclein in the brain [10]. However, the exact cellular mechanisms that cause selective neuronal death in PD are still not fully understood yet [11].

According to PubMed, research on ER stress in PD began in 2000 and gradually increased. Since 2016, more than 46 papers have been intensively published every year (Figure 3).

The following papers are organized by year of publication in PubMed: 6-Hydroxydopamine (6-OHDA) and 1-methyl-4-phenyl-pyridinium (MPP+) are widely used to induce dopaminergic neuron death in in vitro and in vivo PD models. An association between ER stress and the UPR has been reported in neuronal PC12 cells [12], sympathetic neurons from PERK null mice [13], and the MN9D cells’ dopaminergic cell line [13]. ER-localized molecular chaperones (GRP78/BiP and GRP94) have been revealed in SH-SY5Y cells [14]. The relationship between Parkin protein and ER stress in PD has also been proven [15]. IRE1alpha, an ER stress molecule, participates in unfolded protein responses through its interaction with the Jun activation domain-binding protein-1 (JAB1) [16]. Alpha-synuclein cytotoxicity is related to ER stress in PD [17,18,19,20,21]. The ER stress molecules CHOP/Gadd153 and GRP78/BiP are increased in human neuroblastoma SH-SY5Y cells treated with 6-OHDA, indicating that ER dysfunction is involved in the mechanisms induced by 6-OHDA in SH-SY5Y cells [22]. The relationship between alpha-synuclein and PD concerning ER stress has been reviewed [23,24,25]. As ER stress markers, the phosphorylated pancreatic ER kinase (p-PERK) and phosphorylated eukaryotic initiation factor 2alpha (p-eIF2alpha) have been detected in the substantia nigra of PD patients, suggesting that the UPR’s activation is closely related to the accumulation of alpha-synuclein aggregates [26]. The emerging mechanisms of ER stress in PD have been reviewed [27,28] and demonstrated [29].

Parkin has a role in the crosstalk between ER and mitochondrial stress, suggesting that Parkin has a role in the crosstalk between ER and mitochondrial stress, suggesting that both ER and mitochondrial stress can contribute to the pathogenesis of PD [30]. GRP78/BiP can bind to α-synuclein and increase the aggregated α-synuclein’s accumulation in in vitro and in vivo models, indicating that the activation of the UPR pathway in the PD brain is associated with α-synuclein’s accumulation occurring, in part, within the ER [31]. Calì et al. [32] have reviewed the interplay between the mitochondria, ER, and proteasome systems in PD-associated neurodegeneration. When molecular events occurring during ER stress and the unfolded protein response are evaluated, it has been found that the ER stress response plays a role in neurodegenerative disorders, including AD, PD, ALS, and prion diseases [33,34,35,36,37]. Alpha-synuclein can inhibit ATF6, a protective branch of the UPR, suggesting a link between ER stress and the role of the UPR in PD [38]. Tsujii et al. have reviewed the involvement of ER stress with in vitro and in vivo PD models [39]. GRP78/BiP and p-PERK are increased in the Lewy bodies of patients with dementia [40]. An increase in activating transcription factor 4 (ATF4) as a member of the PERK signaling pathway has been identified in experimental animal models and the postmortem melanin-containing neurons of PD patients [41,42]. The IRE1 pathway can drive PD’s progression by coupling the ER stress [43]. The significant role of ER stress in PD pathogenesis has been highlighted [44]. eIF2α phosphorylation mediated by PERK can protect DA neurons against chronic heat stress in Drosophila [45]. The neurodegenerative disease can be activated or inhibited through the PERK pathway [46].

Therefore, the target molecules of ER stress for PD are GRP78/BiP, CHOP, PERK, IRE1, ATF6, and ATF4, suggesting that these ER stress molecules can act mainly as pathological factors in PD.

### 2.2. ER Stress and Alzheimer’s Disease (AD)

AD is a neurodegenerative disease that can lead to progressive impairments in memory, behavioral, and thinking functions. AD causes the loss of neurons in several brain regions within the prefrontal cortex, hippocampus, and basal forebrain area [47]. Pathological features of AD patients’ brain tissues include extracellular senile plaques with amyloid β-peptide (Aβ) deposits and intracellular accumulation of the tau protein with neurofibrillary tangles (NFTs). AD includes early-onset familial AD (FAD) and later-onset sporadic AD. Mutations in presenilin PS1 and PS2 are associated with FAD [47].

According to PubMed, research on ER stress in AD was started in 1997. The number of such research studies then gradually increased. Since 2016, more than 50 papers on ER stress in AD have been published intensively annually (Figure 4).

The following papers are organized according to the year presented in PubMed: For the pathogenesis of neurodegeneration in AD, presenilin has been suggested to be related to ER [48,49]. PS1 mutations or loss of the PS1 function can affect the UPR, such as the expression of GRP78/BiP or CHOP, suggesting that PS1 can increase vulnerability to ER stress in association with the UPR [50,51]. Activation of the UPR in AD and increased GRP78/BiP and p-PERK in normal neurons suggest that the UPR is involved in AD neurodegeneration [52]. ER stress can be a critical trigger in the neuronal response to inflammatory activity. This has been reviewed in terms of the pathology of AD [53].

Aβs can activate the ER stress response factor X-box binding protein 1 (XBP-1) in transgenic flies and mammalian-cultured neurons [54]. AD pathogenesis involves ER stress and hyperphosphorylation of the tau protein [55]. Although the oligomeric form of Aβs has been thought to play a critical role in AD, the mechanisms involved remain unclear in the pathogenesis of AD. Aβs can induce ER stress in AD, suggesting that targeting this toxic component can enable the treatment of various neurodegenerative diseases [56]. Differential manifestations of ER stress and docosahexaenoic acid’s (DHA’s) reactiveness may help explain the variable clinical results obtained with DHA treatment, suggesting that DHA might be adequate for a subset of AD patients [57]. The IgG Fcγ receptor II-b (FcγRIIb) can mediate Aβ neurotoxicity and neurodegeneration. Soluble Aβ oligomers can interact with FcγRIIb in vitro and in AD brains to activate ER stress and caspase-12 [58]. The pharmacological inhibition of brain inflammation and ER stress in mice can prevent glucose intolerance, demonstrating that AD-associated Aβ oligomers can affect the central nervous system [59,60]. This result suggests a novel molecular mechanism between metabolic disease and hypothalamic dysfunction in AD. The complex roles of ER stress, including XBP1 and PERK, during Aβ pathogenesis, indicate the status of disease progression [61]. Inflammasome-mediated peripheral inflammation can contribute to AD pathology [62]. ER stress and the UPR are critical to AD development and onset pathogenesis [63,64]. ER stress and the UPR in cellular aging and neuroinflammatory processes can lead to memory impairment and synapse dysfunction in AD [65].

Therefore, the target molecules of ER stress for AD are GRP78/BiP, CHOP, PERK, and XBP-1, suggesting that these ER stress molecules can act mainly as pathological factors in AD.

### 2.3. ER Stress and Amyotrophic Lateral Sclerosis (ALS)

ALS is a progressive neurodegenerative disease of neurons in the spinal cord and the brain, resulting in a loss of muscle control. Motor neuron loss continues until the person loses the ability to eat, speak, move, and breathe. ALS eventually causes paralysis and premature death due to respiratory failure [66,67,68]. ALS is most common in people aged 40 to 70 years, regardless of racial or ethnic group, although it can occur at younger ages. ALS has two types: sporadic ALS and familial ALS. Sporadic ALS is the most common form in the United States, accounting for approximately 90 to 95% of all cases. It occurs randomly without a known cause or family history. Familial ALS affects 5 to 10% of people and is inherited [69,70]. Although the disease usually progresses rapidly, the underlying causes of neuronal death are not fully known.

According to PubMed, research on ER stress in ALS started in 2003 and gradually increased. Since 2013, more than 20 papers have been published yearly (Figure 5).

The following papers are organized according to the year presented in PubMed: Motor neurons in the spinal cords of transgenic mice with the mutant SOD1 linked to familial amyotrophic local sclerosis (FALS) show a significant increase in GRP78/BiP just before motor symptoms begin, suggesting that ER stress is significantly related to the pathogenesis of FALS with SOD1 mutation [71]. When caspase-12 activation occurs in the spinal cord of a transformed ALS mouse, oxidation and ER-induced stress are induced, resulting in neuronal cell death and disease progression [72]. This suggests that caspase-12 and the ER stress pathways for neuronal apoptosis might be potential new targets for ALS treatment. The balance between the anti- and pro-apoptotic proteins associated with ER stress can induce damage by affecting the presymptomatic phase in an ALS mouse model, suggesting that this imbalance might be involved in the pathological cause of motor neuron degeneration in ALS [73]. Alterations in the fatty acid composition, mitochondrial function, and proteasome activity can be induced by excitotoxicity in the neurons, thereby inducing oxidative stress and, finally, endoplasmic reticulum stress in sporadic ALS [74]. The BH3-only protein, Puma, associated with ER stress, is critically involved in neurodegenerative disease progression in SOD1G93A mice, indicating that Puma is an essential factor in controlling chronic neurodegeneration in ALS and other neurodegenerative disorders, including defects in the protein quality control [75]. SOD1-induced protein misfolding associated with ALS mutations can induce ER stress, suggesting that it might contribute to motor neuron degeneration in ALS pathogenesis [76].

The interaction between Derlin-1 and SOD1 mutation can induce ER stress. Subsequent ASK1 activation is critical for disease progression in familial ALS [77]. Vesicle-associated membrane protein-associated protein B (VAPB), a novel ALS causative gene, plays a pivotal role in the UPR to ER stress. The ALS-linked P56S mutation in VAPB can abolish the function of VAPB, resulting in motor neuron vulnerability to ER stress [78]. A review article has presented ER stress and UPR dysfunction concerning ALS pathological mechanisms [79]. Although all motoneurons in ALS can be preferentially affected, the ER stress response in the specific motoneuron subtypes of ALS can affect the gradual onset of weakness and paralysis [80]. A new function of XBP-1 in autophagy indicates the critical correlation between the increased autophagy in the motoneurons and a reduced accumulation of mutant SOD1 aggregates, which can protect the neurons against neurodegeneration in ALS [81,82]. The upregulation of CHOP in the motor neurons and glial cells might play a pivotal role in the pathogenesis of ALS [83]. ER stress may act as a possible risk factor for the development of ALS by increasing the susceptibility of the wild-type SOD1 to aggregation during aging [84].

Therefore, the target molecules of ER stress for ALS are GRP78/BiP, CHOP, and XBP-1, suggesting that these ER stress molecules can act mainly as pathological factors in ALS.

### 2.4. ER Stress and Transmissible Spongiform Encephalopathies (TSEs)

TSEs or prion diseases are rare progressive neurodegenerative brain disorders affecting humans and animals with a spongy form in the brain. TSEs include Creutzfeldt–Jakob disease, bovine spongiform encephalopathy (BSE), and scrapie. The pathological hallmark of TSEs is the accumulation of a misfolded protease-resistant form of prion protein (PrP) in the cerebrum, leading to extensive neuronal cell death as the disease progresses [85,86,87].

According to PubMed, ER stress in TSE was not studied much until recently. However, in 2000, papers began to be published gradually. In 2015, 12 papers were published. The number of published papers gradually decreased until recently (Figure 6).

The following articles are organized by year of publication in PubMed: Caspase-12 is increased in cells showing misfolded prion protein (PrP^Sc^). It is also increased in the brain tissues of PrPSc-infected mice and Creutzfeldt–Jakob disease patients [88]. This is the first study to present that ER stress and the activation of caspase-12 can cause neuronal cell death by accumulating mutant prion protein in the neurons. The overexpression of GRP58 can protect the cells from PrP^Sc^ toxicity and reduce caspase-12, suggesting that the expression of GRP58 can act as a neuroprotective factor against prion neurotoxicity [89]. Studies on GRP58 have suggested that it can be used to develop new targets for treating prion diseases [89]. The induction of ER stress in transgenic mice expressing PrP variants is accompanied by reduced translocation, a finding that could link age-dependent clinical and histological signs to PrP-mediated neurodegeneration [90]. One mechanism of prion-mediated neurodegeneration involves indirect ER stress-dependent effects on early PrP biosynthesis and metabolism [90]. The role of GRP78/BiP has been demonstrated to be important in prion diseases through in vivo and in vitro approaches [91]. The upregulation of PERK, GRP78/BiP, the ER protein, disulfide isomerase, and ubiquitin in prion disease indicates that ER stress and proteasome damage can initiate the early stages of spontaneous prion disease [92].

Therefore, the target molecules of ER stress for TSEs are GRP58, GRP78/BiP, and PERK, suggesting that these ER stress molecules can act mainly as pathological factors in TSEs.

### 2.5. ER Stress and Polyglutamine Diseases

Polyglutamine diseases are manifested by progressive neurodegeneration that can lead to behavioral and physical impairments. Polyglutamine diseases include Huntington’s disease (HD), dentatorubral-pallidoluysian atrophy, spinal and bulbar muscular atrophy, and spinocerebellar ataxias 1, 2, 3, 6, 7, and 17 [93]. The disease features selective neuronal cell death and the accumulation of intracellular protein aggregates in cultured cells, transgenic animals, and human postmortem brain tissues. The pathophysiology of polyglutamine disorders with polyglutamine expansion and protein deposits remains unclear [94]. ASK1 is essential for neuropathological changes in polyglutamine diseases. It is crucial for killing neurons by inducing ER stress [95]. In HD, the N-terminal huntingtin protein can increase ER stress-related GRP78/BiP, CHOP, c-Jun-N-terminal kinase (JNK) phosphorylation, and caspase-12 activation [96]. Polyglutamine-expanded huntingtin fragments expressing yeast cells and neuron-like PC12 cells can impair the ER-associated degradation (ERAD) proteins Npl4, Ufd1, and p97 [97]. Soluble oligomeric polyglutamine-expanded huntingtin can induce ER stress before its aggregation through p97/VCP [98].

Therefore, the target molecules of ER stress for polyglutamine diseases are ASK1, GRP78/BiP, CHOP, and ERAD, suggesting that these ER stress molecules can act mainly as pathological factors in polyglutamine diseases.

### 2.6. ER Stress and Neuronal Storage Diseases or Lysosomal Storage Disease (LSD)

Lysosomal storage disease (LSD) is a hereditary metabolic disorder with a defective lysosomal function. LSD includes more than 70 diseases with an abnormal lysosomal function, most of which are inherited as autosomal recessive traits. Although these disorders are rare individually, collectively, they occur in one in five thousand births [99]. Most LSDs show clinically progressive neurodegeneration, although symptoms of other organ systems are also frequently present. LSD has been classified and described in considerable detail previously [99].

Pelizaeus–Merzbacher disease (PMD) is an X-linked recessive pediatric and neurodegenerative disorder characterized by diffuse hypomyelination of the central nervous system. Mutations, duplications, or deletions of the phospholipid protein (PLP) gene can cause PMD by accumulating the PLP and/or DM20 proteins in the ER. CHOP in oligodendrocytes is directly involved in mice with the gene deletion causing PMD [100]. The accumulation of GM2-gangliosidosis in a mouse model of Sandhoff disease is caused by reduced calcium uptake into the ER due to decreased sarcoplasmic/ER calcium ATPase (SERCA), resulting in neuronal death [101]. GM1-ganglioside (GM1) is a significant sialoglycolipid in nerve cell membranes that can regulate calcium homeostasis. In a mouse model with GM1-gangliosidosis, the upregulation of GRP78/BiP and CHOP and the activation of JNK2 and caspase-12 can induce neuronal cell death, suggesting that the UPR can induce the accumulation of the sialoglylipid GM1 in the neuronal cells and then cause neuronal apoptosis [102]. CLN6 is a non-glycosylated ER-resident membrane protein with an unknown function. Mutant Cln6 can prevent the accumulation of the misfolded Cln6 protein by proteasomal degradation, impair constitutive autophagy by lysosomal dysfunction, and promote neurodegeneration [103]. A deficiency in the hexosaminidase activity can disrupt the normal metabolism of GM2-ganglioside, leading to progressive neurodegenerative diseases [104]. In cultured neurons, GM2-ganglioside accumulation in the ER can induce luminal Ca^2+^ depletion, activating PERK. Therefore, it can induce neurite atrophy and apoptosis [104]. Pelizaeus–Merzbacher-like disease type 1 (PMLD1) is a hypomyelination disorder in patients with mutations in the GJC2 coding for Connexin47 (Cx47). PMLD1 can cause nystagmus, cerebellar ataxia, convulsions, and changes in the CNS’s white matter. In three PMLD1-related mutants (p.P87S, p.Y269D, and p.M283T), Cx47 can induce ER stress, including the UPR, and activate the apoptotic pathway [105].

Therefore, the target molecules of ER stress for LSD are GRP78/BiP, CHOP, PERK, IRE1, ATF6, and ATF4, suggesting that these ER stress molecules can act mainly as pathological factors in LSD.

### 2.7. ER Stress and Acute Neurodegeneration

ER stress can induce acute brain disorders such as cerebral ischemia and traumatic brain injury, apart from chronic neurodegenerative diseases. Oxygen-regulated protein 150 kD (ORP150) and 94 kDa glucose-regulated protein (GRP94) as ER chaperones can be directly involved in the ER response in brain injury. ORP150 is elevated in the neurons of a human brain with ischemic stress [106]. In human neuroblastoma cells exposed to hypoxia/reoxygenation, GRP94 is increased [107]. Caspase-12-mediated ER apoptosis might be involved in rat traumatic brain injury pathology [108]. Primary hippocampal neurons in CHOP-deficient mice are resistant to hypoxia-reoxygenation-induced apoptosis, indicating that ischemia-induced hippocampal neuronal death is caused by the ER stress pathway associated with CHOP induction [109]. Global cerebral ischemia in transgenic rats can increase the expression of ATF4 and CHOP, indicating that superoxide is involved in ER stress-induced cell death [110].

Therefore, the target molecules of ER stress for acute neurodegeneration are GRP94, CHOP, and ATF4, suggesting that these ER stress molecules can act mainly as pathological factors in acute neurodegeneration.

## 3. Melatonin

Melatonin is a potent antioxidant and free radical scavenger [111,112] and protects against inflammation, apoptosis, or autophagy in physiological and pathophysiological conditions [113,114,115,116]. In addition, melatonin modulates ER stress and UPR dysfunction in cancers, liver diseases, and other pathologies [115,116,117]. In particular, review articles introduce that melatonin can suppress various diseases related to ER stress: chronic intestinal inflammation and colon cancer [118], breast cancer [119], osteoarthritis [120], hepatocellular carcinoma [121], delaying ovarian aging [122], acute myocardial infarction [123], and diabetic cardiomyopathy [124].

Although there are many papers on the prevention or suppression of diseases by melatonin, there are very few papers about the effects of melatonin on ER stress in neurons and neurodegenerative diseases. The following describes the effects of melatonin reported so far, focusing on its effects on neurons and neurodegenerative diseases related to ER stress.

### 3.1. Melatonin and ER Stress in Neurons

Melatonin can significantly reduce the neuron splicing of the XBP-1 mRNA, increase eIF2α phosphorylation, and elevate the expression of the chaperone proteins GRP78/BiP and Hsp70 in hypoxia-ischemia of a rat brain [125]. In addition, melatonin can significantly reduce CHOP expression. Melatonin can reduce ER stress induced by neonatal hypoxia-ischemia in newborn rats through SIRT-1 as the modulation and neuroprotection [125]. Melatonin can protect neuroblastoma cells against methamphetamine-induced ER stress and apoptosis by modulating CHOP, spliced XBP1, caspase-12, and caspase-3 [126]. Methamphetamine can induce ER stress in glial cells by stimulating the UPR to increase PERK phosphorylation, ATF6 expression, and the phosphorylated inositol-requiring enzyme 1 (p-IRE1). Moreover, the expression levels of GRP78/BiP and CHOP, caspase-12 activation, eIF2α phosphorylation, and XBP-1 mRNA expression are increased. However, melatonin can reduce ER stress through methamphetamine toxicity by reducing the expression of the ER stress response genes and proteins [127]. Ischemia/reperfusion in acute neuronal injury after an ischemic stroke can induce neuronal apoptosis by increasing ER stress, including the phosphorylation of PERK and IRE1 and the expression of ATF6 and CHOP. Melatonin pretreatment can attenuate ischemia/reperfusion-induced ER stress [128]. Melatonin treatment can attenuate the insulin-induced mRNA or protein expression levels of ER stress markers such as p-eIF2α, ATF4, CHOP, sXBP1, p-IRE1, and p-ASK1. Therefore, melatonin can regulate neuronal cell death induced by ER stress under insulin resistance by reducing the ER stress in SH-SY5Y neuroblastoma cells [129].

Melatonin can effectively downregulate the levels of ER stress molecules GRP78/BiP, CHOP, and caspase-12 proteins in kainic acid-induced N2a cells [130]. Melatonin can also inhibit tau hyperphosphorylation, the phosphorylation of PERK and eIF2α, and the expression levels of ATF4 and GRP78/BiP in the kainic acid-treated mouse hippocampus [131]. Melatonin can attenuate ER stress, such as the phosphorylation of PERK and eIF2α and the expression levels of ATF4 and CHOP in neurons exposed to oxygen and glucose deprivation and in rats subjected to transient focal cerebral ischemia [132]. These results indicate that melatonin suppresses post-ischemic ER stress after an ischemic stroke. The administration of melatonin can significantly decrease the mRNA and protein levels of ATF6 and CHOP in intracerebral hemorrhage rats, indicating that melatonin can protect neurons against apoptosis by suppressing ATF6 and CHOP [133]. Melatonin can decrease ER stress, such as the p-IRE1, p-PERK, GRP78/BiP, and CHOP, in the chronic cerebral hypoperfusion of rats, suggesting that melatonin can improve cognitive impairment following the induction of bilateral typical carotid artery occlusion by attenuating AD markers and reducing ER stress [134].

Therefore, these studies show that the first target molecule of ER stress for melatonin is CHOP, and the second target molecules are PERK and GRP78/BiP. Therefore, melatonin is a crucial strategy to protect and treat neurons against ER stress.

### 3.2. Melatonin as ER Stress Modulator against Neurodegenerative Diseases

Direct oxidative damage and organelle lesions in rats can increase the expression of ER stress-related proteins, including GRP78/BiP and CHOP. The addition of melatonin can partially suppress ER stress-related behavioral and molecular disturbances. These results suggest that melatonin is directly related to AD-like behavioral and molecular pathologies with ER stress-related mechanisms [135]. One review article has presented the effects of melatonin on the fundamental ER stress mechanisms, focusing on its ability to modulate autophagy and apoptotic processes in the development of cancer cells, neurodegeneration, liver disease, and other pathologies [117].

Melatonin can protect mesenchymal stem cells from ischemic damage by inhibiting ER stress (such as decreasing the phosphorylation of PERK/eIF2α/IRE1α and the expression levels of ATF4 and CHOP proteins) and autophagy both in vivo and in vitro. In addition, melatonin can increase the cellular prion protein (PrPC) expression to prevent apoptotic cell death induced by ER stress and autophagy [136]. Melatonin is an endogenously produced molecule that can act as a copper chelator, a potent antioxidant, and an inhibitor of ER stress and the UPR in the liver and brain. Therefore, melatonin can potentially lower copper levels and limit the progression of Wilson’s disease [137]. Melatonin can reduce Aβ peptide accumulation and ER stress markers such as the protein levels of GRP78/BiP, CHOP, and caspase-12 in an AlCl_3_-treated AD model. Therefore, adding melatonin might be an alternative way to alleviate the onset of Alzheimer’s disease [138].

Therefore, an in vivo study of melatonin has also revealed that the molecules of ER stress are CHOP, GRP78/BiP, and PERK. Therefore, melatonin has a significant strategic value in inhibiting and treating neurodegeneration caused by ER stress.

## 4. Conclusions

ER participates in various cellular processes, from which proteins are synthesized and transported to their final destinations after correct processing and modification. In neurodegenerative diseases, misprocessing occurs in many cellular processes related to ER, which can act as ER stress and finally cause neuronal cell death. However, two questions remain in this process: (1) how much cellular degeneration will occur due to ER stress? and (2) how much would another pathway induce cellular degeneration? The proteins that regulate or participate in the ER–mitochondrial interaction that causes cell death are not fully understood yet. ER stress molecules are involved in apoptosis, leading to a loss of ER function and the activation of the apoptosis cascade. Likewise, neuronal death occurs when proteins associated with neurodegenerative diseases accumulate to form cellular aggregates. This probably depends on the nature and function of the protein in question. Therefore, we think it is necessary to clearly understand and solve the mutual relationship between the molecules involved in ER stress and neuronal cell death.

Taken together, the reviewed findings suggest that we can target specific stages of the disease by focusing on the ER stress target molecules of melatonin to prevent or treat neurodegenerative diseases. This review suggests that CHOP, GRP78/BiP, and PERK are the ER stress target molecules of melatonin. Of course, more research studies are needed to test this possibility.

## Figures and Tables

**Figure 1 ijms-24-02381-f001:**
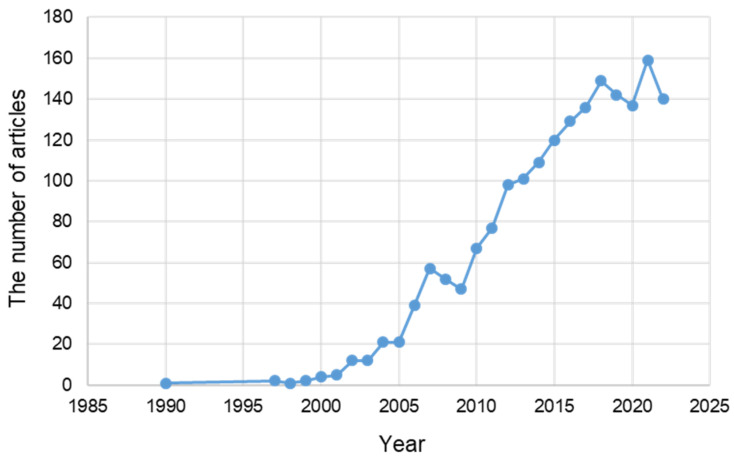
The number of articles related to ER stress among articles on neurodegenerative diseases published in PubMed from 1990 to 23 November 2022.

**Figure 2 ijms-24-02381-f002:**
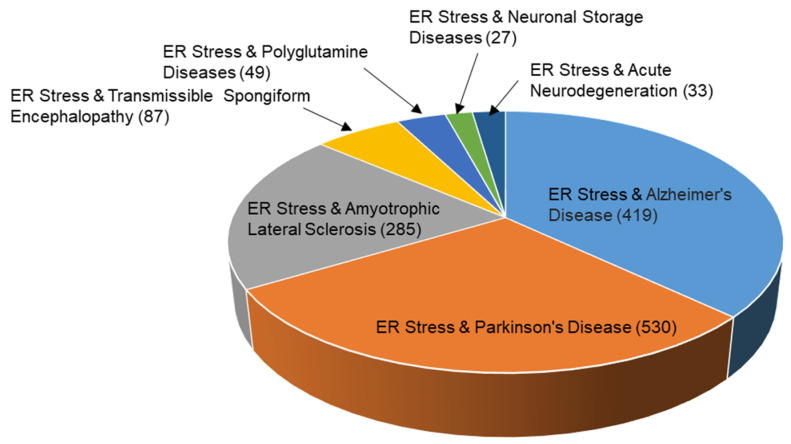
The number of articles related to neurodegenerative diseases and ER stress investigated by PubMed by 23 November 2022.

**Figure 3 ijms-24-02381-f003:**
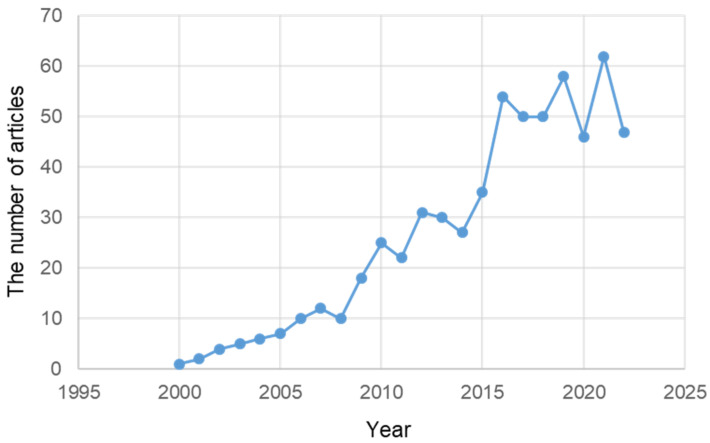
The number of articles related to ER stress in PD published in PubMed from 2000 to 23 November 2022.

**Figure 4 ijms-24-02381-f004:**
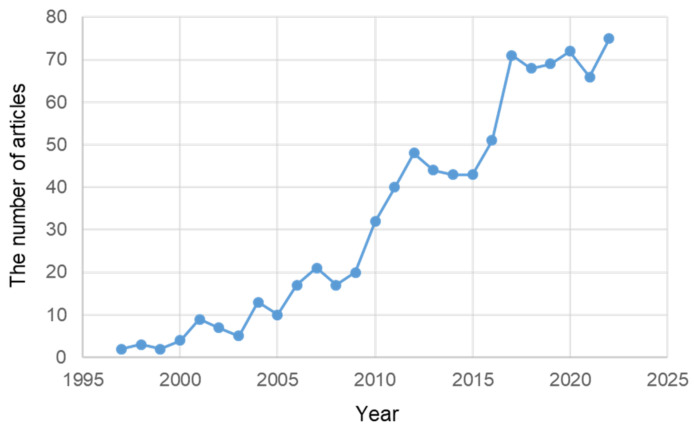
The number of articles related to ER stress in AD published in PubMed from 1997 to 23 November 2022.

**Figure 5 ijms-24-02381-f005:**
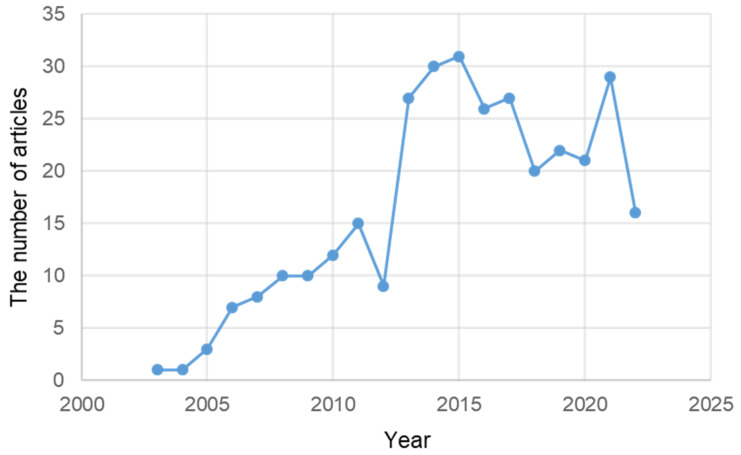
The number of articles related to ER stress in ALS published in PubMed from 2003 to 23 November 2022.

**Figure 6 ijms-24-02381-f006:**
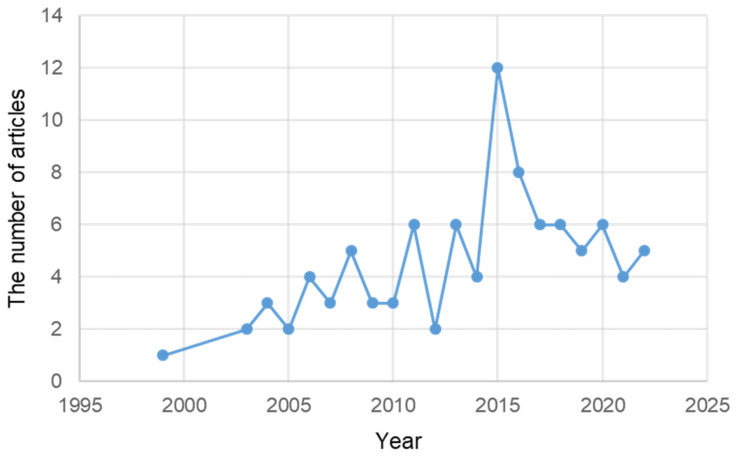
The number of articles related to ER stress in TSE published in PubMed from 1999 to 23 November 2022.

## Data Availability

Not applicable.

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
