# Peer review of "Melatonin Can Modulate Neurodegenerative Diseases by Regulating Endoplasmic Reticulum Stress"

_ijms, 2023, doi:10.3390/ijms24032381_

Round 1
Reviewer 1 Report
The manuscript summarizes the current knowledge on melatonin modulation in neurodegeneration through ER stress. The general topic is interesting and the manuscript collected several studies in the field. I would recommend publishing the manuscript only after two main alterations listed below.
- English language and style requires editing and it is advised to get a native speaker revision; some sentences are too long, some verbs are not well used, some sentences are not well understood, repeating some terms, etc.
- Where many studies are mentioned regarding each subject, some connections and then conclusions need to be written.
-If possible, add more recent papers specifically from 2022.
Author Response
January 18, 2023
Dear Reviewer 1,
Thank you for your letter and for the reviewers’ comments concerning our manuscript entitled " Melatonin can modulate neurodegenerative diseases by regulating endoplasmic reticulum stress."
We completely agree with your suggestions regarding our manuscript. The manuscript is completely revised as you and your colleague requested.
The correction and revisions are as follows:
The manuscript summarizes the current knowledge on melatonin modulation in neurodegeneration through ER stress. The general topic is interesting and the manuscript collected several studies in the field. I would recommend publishing the manuscript only after two main alterations listed below.
- English language and style require editing and it is advised to get a native speaker revision; some sentences are too long, some verbs are not well used, some sentences are not well understood, repeating some terms, etc.
Answer: Thank you very much for your good comments.
I received English proofreading through a proofreading company (Harrisco). The Certificate of Editing is attached below.
I am very sorry if you think that the reviewer must have been under considerable stress while reviewing the thesis. However, we also feel that it is too insufficient even though we have received the maximum correction through the correction company. However, since we did our best, we are lacking, but we hope you will understand.
- Where many studies are mentioned regarding each subject, some connections and then conclusions need to be written.
Answer: Thank you very much for your good comments.
Since it was described by year, it was not possible to draw a conclusion, so it was simply summarized and added focusing on ER stress molecules.
Lines 116-118: Therefore, the target molecules of ER stress for PD are GRP78/BiP, CHOP, PERK, IRE1, ATF6, and ATF4, suggesting that these ER stress molecules can act mainly as pathological factors in PD.
Lines 160-162: Therefore, the target molecules of ER stress for AD are GRP78/BiP, CHOP, PERK, and XBP-1, suggesting that these ER stress molecules can act mainly as pathological factors in AD.
Lines 216-217: Therefore, the target molecules of ER stress for ALS are GRP78/BiP, CHOP, and XBP-1, suggesting that these ER stress molecules can act mainly as pathological factors in ALS.
Lines 251-253: Therefore, the target molecules of ER stress for TSEs are GRP58, GRP78/BiP, and PERK, suggesting that these ER stress molecules can act mainly as pathological factors in TSEs.
Lines 270-272: Therefore, the target molecules of ER stress for polyglutamine diseases are ASK1, GRP78/BiP, CHOP, and ERAD, suggesting that these ER stress molecules can act mainly as pathological factors in polyglutamine diseases.
Lines 305-307: Therefore, the target molecules of ER stress for LSD are GRP78/BiP, CHOP, PERK, IRE1, ATF6, and ATF4, suggesting that these ER stress molecules can act mainly as pathological factors in LSD.
Lines 321-323: Therefore, the target molecules of ER stress for acute neurodegeneration are GRP94, CHOP, and ATF4, suggesting that these ER stress molecules can act mainly as pathological factors in acute neurodegeneration.
Lines 325-332: Melatonin is a potent antioxidant and free radical scavenger [111, 112] and pro-tects inflammation, apoptosis, or autophagy in physiological and pathophysiological conditions [113-116]. In addition, melatonin modulates ER stress and UPR dysfunction in cancers, liver diseases, and other pathologies [115-117]. In particular, review articles introduce that melatonin can suppress various diseases related to ER stress: chronic intestinal inflammation and colon cancer [118], breast cancer [119], osteoarthritis [120], hepatocellular carcinoma [211], delaying ovarian aging [122], acute myocardial infarction [123], and diabetic cardiomyopathy [124].
Lines 701-733:
- Bonnefont-Rousselot, D.; Collin, F.; Jore, D.; Gardès-Albert M. Reaction mechanism of melatonin oxidation by reactive oxygen species in vitro. Pineal Res. 2011, 50, 328-335. doi: 10.1111/j.1600-079X.2010.00847.x.
- Galano, A.; Tan, D.X.; Reiter, R.J. Melatonin as a natural ally against oxidative stress: a physicochemical examination. Pineal Res. 2011, 51, 1-16. doi: 10.1111/j.1600-079X.2011.00916.x.
- Kim, S.H.; Lee, S.M. Cytoprotective effects of melatonin against necrosis and apoptosis induced by ischemia/reperfusion injury in rat liver. Pineal Res. 2008, 44, 165-171. doi: 10.1111/j.1600-079X.2007.00504.x.
- Muñoz-Casares, F.C.; Padillo, F.J.; Briceño, J.; Collado, J.A.; Muñoz-Castañeda, J.R.; Ortega, R.; Cruz, A.; Túnez, I.; Montilla, P.; Pera, C.; Muntané, J. Melatonin reduces apoptosis and necrosis induced by ischemia/reperfusion injury of the pancreas. Pineal Res. 2006, 40, 195-203. doi: 10.1111/j.1600-079X.2005.00291.x.
- Tuñón, M.J.; San-Miguel, B.; Crespo, I.; Laliena, A.; Vallejo, D.; Álvarez, M.; Prieto, J.; González-Gallego, J. Melatonin treatment reduces endoplasmic reticulum stress and modulates the unfolded protein response in rabbits with lethal fulminant hepatitis of viral origin. J Pineal Res. 2013, 55, 221-228. doi: 10.1111/jpi.12063.
- San-Miguel, B.; Crespo, I.; Vallejo, D.; Álvarez, M.; Prieto, J.; González-Gallego, J.; Tuñón, M.J. Melatonin modulates the autophagic response in acute liver failure induced by the rabbit hemorrhagic disease virus. Pineal Res. 2014, 56, 313-321. doi: 10.1111/jpi.12124.
- Fernández, A.; Ordóñez, R.; Reiter, R.J.; González-Gallego, J.; Mauriz, J.L. Melatonin and endoplasmic reticulum stress: relation to autophagy and apoptosis. Pineal Res. 2015, 59, 292-307. doi: 10.1111/jpi.12264.
- Motilva, V.; García-Mauriño, S.; Talero, E.; Illanes, M. New paradigms in chronic intestinal inflammation and colon cancer: role of melatonin. Pineal Res. 2011, 51, 44-60. doi: 10.1111/j.1600-079X.2011.00915.x.
- Naziroğlu, M.; Tokat, S.; Demirci, S. Role of melatonin on electromagnetic radiation-induced oxidative stress and Ca2+ signaling molecular pathways in breast cancer. Recept. Signal. Transduct Res. 2012, 32, 290-297. doi: 10.3109/10799893.2012.737002.
- Hosseinzadeh, A.; Kamrava, S.K.; Joghataei, M.T.; Darabi, R.; Shakeri-Zadeh, A.; Shahriari, M.; Reiter, R.J.; Ghaznavi, H.; Mehrzadi, S. Apoptosis signaling pathways in osteoarthritis and possible protective role of melatonin. Pineal Res. 2016, 61, 411-425. doi: 10.1111/jpi.12362.
- Mortezaee, K. Human hepatocellular carcinoma: Protection by melatonin. Cell. Physiol. 2018, 233, 6486-6508. doi: 10.1002/jcp.26586.
- Yang, Y.; Cheung, H.H.; Zhang, C.; Wu, J.; Chan, W.Y. Melatonin as Potential Targets for Delaying Ovarian Aging. Drug Targets 2019, 20, 16-28. doi: 10.2174/1389450119666180828144843.
- Fu, Z.; Jiao, Y.; Wang, J.; Zhang, Y.; Shen, M.; Reiter, R.J.; Xi, Q.; Chen, Y. Cardioprotective Role of Melatonin in Acute Myocardial Infarction. Physiol. 2020, 11, 366. doi: 10.3389/fphys.2020.00366.
- Huang, K.; Luo, X.; Zhong, Y.; Deng, L.; Feng, J. New insights into the role of melatonin in diabetic cardiomyopathy. Res. Perspect. 2022, 10, e00904. doi: 10.1002/prp2.904.Melatonin
-If possible, add more recent papers specifically from 2022.
Answer: Thank you very much for your good comments.
We tried to add papers published as recently as possible. However, if you see the reviewer as lacking, I am very sorry. However, since we did our best, we are lacking, but we hope you will understand.
**All of the edited sections and references were changed with the blue words.
***The Certificate of Editing is attached.
I hope that the revised manuscript is now acceptable for publication in the IJMS. We are looking forward to receiving your answer soon.
Sincerely,
Yeong-Min Yoo Ph.D
College of Life Science,
Gangneung-Wonju National University,
Gangneung, Gangwon-do 25457, Republic of Korea
Email: yyeongm@hanmail.net

Reviewer 2 Report
This review article described the unfolded protein response (UPR) in neurodegenerative diseases, but how the UPR activity promoted neurodegenerative diseases were skipped. It damaged the strength of this paper.
Author Response
January 18, 2023
Dear Reviewer 2,
Thank you for your letter and for the reviewers’ comments concerning our manuscript entitled " Melatonin can modulate neurodegenerative diseases by regulating endoplasmic reticulum stress."
We completely agree with your suggestions regarding our manuscript. The manuscript is completely revised as you and your colleague requested.
The correction and revisions are as follows:
The review article is well organized, but repetitive and not novel. There are several already well written reviews on Neurodegeneration-ER-Melatonin interlinks. One thing that can further increase the value of this work is actually citing the original researches in your review instead of other reviews. Further, including schematic diagrams/graphics/flowcharts/other visual media for section 2.1 through 3.2. There is good amount of text but visual media makes it easy of the readers, especially new researchers who would refer your article in the future.
The most important point: The majority of this review is focused of ER and neurodegeneration/neuroinflammation. There is very less information and focus on melatonin. I would suggest adding more, updated and latest information and evidences on the role of melatonin as well as a proposed solution and mechanistic/experimental approaches to what can be done in future to collect more evidence in line with your review, especially conclusion.
Answer: Thank you very much for your good comments.
We tried to add papers published as recently as possible. However, if you see the reviewer as lacking, I am very sorry. However, since we did our best, we are lacking, but we hope you will understand.
Since it was described by year, it was not possible to draw a conclusion, so it was simply summarized and added focusing on ER stress molecules.
Lines 116-118: Therefore, the target molecules of ER stress for PD are GRP78/BiP, CHOP, PERK, IRE1, ATF6, and ATF4, suggesting that these ER stress molecules can act mainly as pathological factors in PD.
Lines 160-162: Therefore, the target molecules of ER stress for AD are GRP78/BiP, CHOP, PERK, and XBP-1, suggesting that these ER stress molecules can act mainly as pathological factors in AD.
Lines 216-217: Therefore, the target molecules of ER stress for ALS are GRP78/BiP, CHOP, and XBP-1, suggesting that these ER stress molecules can act mainly as pathological factors in ALS.
Lines 251-253: Therefore, the target molecules of ER stress for TSEs are GRP58, GRP78/BiP, and PERK, suggesting that these ER stress molecules can act mainly as pathological factors in TSEs.
Lines 270-272: Therefore, the target molecules of ER stress for polyglutamine diseases are ASK1, GRP78/BiP, CHOP, and ERAD, suggesting that these ER stress molecules can act mainly as pathological factors in polyglutamine diseases.
Lines 305-307: Therefore, the target molecules of ER stress for LSD are GRP78/BiP, CHOP, PERK, IRE1, ATF6, and ATF4, suggesting that these ER stress molecules can act mainly as pathological factors in LSD.
Lines 321-323: Therefore, the target molecules of ER stress for acute neurodegeneration are GRP94, CHOP, and ATF4, suggesting that these ER stress molecules can act mainly as pathological factors in acute neurodegeneration.
Lines 325-332: Melatonin is a potent antioxidant and free radical scavenger [111, 112] and pro-tects inflammation, apoptosis, or autophagy in physiological and pathophysiological conditions [113-116]. In addition, melatonin modulates ER stress and UPR dysfunction in cancers, liver diseases, and other pathologies [115-117]. In particular, review articles introduce that melatonin can suppress various diseases related to ER stress: chronic intestinal inflammation and colon cancer [118], breast cancer [119], osteoarthritis [120], hepatocellular carcinoma [211], delaying ovarian aging [122], acute myocardial infarction [123], and diabetic cardiomyopathy [124].
Lines 701-733:
- Bonnefont-Rousselot, D.; Collin, F.; Jore, D.; Gardès-Albert M. Reaction mechanism of melatonin oxidation by reactive oxygen species in vitro. Pineal Res. 2011, 50, 328-335. doi: 10.1111/j.1600-079X.2010.00847.x.
- Galano, A.; Tan, D.X.; Reiter, R.J. Melatonin as a natural ally against oxidative stress: a physicochemical examination. Pineal Res. 2011, 51, 1-16. doi: 10.1111/j.1600-079X.2011.00916.x.
- Kim, S.H.; Lee, S.M. Cytoprotective effects of melatonin against necrosis and apoptosis induced by ischemia/reperfusion injury in rat liver. Pineal Res. 2008, 44, 165-171. doi: 10.1111/j.1600-079X.2007.00504.x.
- Muñoz-Casares, F.C.; Padillo, F.J.; Briceño, J.; Collado, J.A.; Muñoz-Castañeda, J.R.; Ortega, R.; Cruz, A.; Túnez, I.; Montilla, P.; Pera, C.; Muntané, J. Melatonin reduces apoptosis and necrosis induced by ischemia/reperfusion injury of the pancreas. Pineal Res. 2006, 40, 195-203. doi: 10.1111/j.1600-079X.2005.00291.x.
- Tuñón, M.J.; San-Miguel, B.; Crespo, I.; Laliena, A.; Vallejo, D.; Álvarez, M.; Prieto, J.; González-Gallego, J. Melatonin treatment reduces endoplasmic reticulum stress and modulates the unfolded protein response in rabbits with lethal fulminant hepatitis of viral origin. J Pineal Res. 2013, 55, 221-228. doi: 10.1111/jpi.12063.
- San-Miguel, B.; Crespo, I.; Vallejo, D.; Álvarez, M.; Prieto, J.; González-Gallego, J.; Tuñón, M.J. Melatonin modulates the autophagic response in acute liver failure induced by the rabbit hemorrhagic disease virus. Pineal Res. 2014, 56, 313-321. doi: 10.1111/jpi.12124.
- Fernández, A.; Ordóñez, R.; Reiter, R.J.; González-Gallego, J.; Mauriz, J.L. Melatonin and endoplasmic reticulum stress: relation to autophagy and apoptosis. Pineal Res. 2015, 59, 292-307. doi: 10.1111/jpi.12264.
- Motilva, V.; García-Mauriño, S.; Talero, E.; Illanes, M. New paradigms in chronic intestinal inflammation and colon cancer: role of melatonin. Pineal Res. 2011, 51, 44-60. doi: 10.1111/j.1600-079X.2011.00915.x.
- Naziroğlu, M.; Tokat, S.; Demirci, S. Role of melatonin on electromagnetic radiation-induced oxidative stress and Ca2+ signaling molecular pathways in breast cancer. Recept. Signal. Transduct Res. 2012, 32, 290-297. doi: 10.3109/10799893.2012.737002.
- Hosseinzadeh, A.; Kamrava, S.K.; Joghataei, M.T.; Darabi, R.; Shakeri-Zadeh, A.; Shahriari, M.; Reiter, R.J.; Ghaznavi, H.; Mehrzadi, S. Apoptosis signaling pathways in osteoarthritis and possible protective role of melatonin. Pineal Res. 2016, 61, 411-425. doi: 10.1111/jpi.12362.
- Mortezaee, K. Human hepatocellular carcinoma: Protection by melatonin. Cell. Physiol. 2018, 233, 6486-6508. doi: 10.1002/jcp.26586.
- Yang, Y.; Cheung, H.H.; Zhang, C.; Wu, J.; Chan, W.Y. Melatonin as Potential Targets for Delaying Ovarian Aging. Drug Targets 2019, 20, 16-28. doi: 10.2174/1389450119666180828144843.
- Fu, Z.; Jiao, Y.; Wang, J.; Zhang, Y.; Shen, M.; Reiter, R.J.; Xi, Q.; Chen, Y. Cardioprotective Role of Melatonin in Acute Myocardial Infarction. Physiol. 2020, 11, 366. doi: 10.3389/fphys.2020.00366.
- Huang, K.; Luo, X.; Zhong, Y.; Deng, L.; Feng, J. New insights into the role of melatonin in diabetic cardiomyopathy. Res. Perspect. 2022, 10, e00904. doi: 10.1002/prp2.904.Melatonin
**All of the edited sections and references were changed with the blue words.
***The Certificate of Editing is attached.
I hope that the revised manuscript is now acceptable for publication in the IJMS. We are looking forward to receiving your answer soon.
Sincerely,
Yeong-Min Yoo Ph.D
College of Life Science,
Gangneung-Wonju National University,
Gangneung, Gangwon-do 25457, Republic of Korea
Email: yyeongm@hanmail.net

Reviewer 3 Report
The review article is well organized, but repetitive and not novel. There are several already well written reviews on Neurodegeneration-ER-Melatonin interlinks. One thing that can further increase the value of this work is actually citing the original researches in your review instead of other reviews. Further, including schematic diagrams/graphics/flowcharts/other visual media for section 2.1 through 3.2. There is good amount of text but visual media makes it easy of the readers, especially new researchers who would refer your article in the future.
The most important point: The majority of this review is focused of ER and neurodegeneration/neuroinflammation. There is very less information and focus on melatonin. I would suggest adding more, updated and latest information and evidences on the role of melatonin as well as a proposed solution and mechanistic/experimental approaches to what can be done in future to collect more evidence in line with your review, especially conclusion.
Author Response
January 18, 2023
Dear Reviewer 3,
Thank you for your letter and for the reviewers’ comments concerning our manuscript entitled " Melatonin can modulate neurodegenerative diseases by regulating endoplasmic reticulum stress."
We completely agree with your suggestions regarding our manuscript. The manuscript is completely revised as you and your colleague requested.
The correction and revisions are as follows:
This review article described the unfolded protein response (UPR) in neurodegenerative diseases, but how the UPR activity promoted neurodegenerative diseases were skipped. It damaged the strength of this paper.
Answer: Thank you very much for your good comments.
It is summarized together with the role of melatonin, focusing on ER stress molecules including UPR rather than focusing on UPR among ER.
We tried to add papers published as recently as possible. However, if you see the reviewer as lacking, I am very sorry. However, since we did our best, we are lacking, but we hope you will understand.
Since it was described by year, it was not possible to draw a conclusion, so it was simply summarized and added focusing on ER stress molecules.
Lines 116-118: Therefore, the target molecules of ER stress for PD are GRP78/BiP, CHOP, PERK, IRE1, ATF6, and ATF4, suggesting that these ER stress molecules can act mainly as pathological factors in PD.
Lines 160-162: Therefore, the target molecules of ER stress for AD are GRP78/BiP, CHOP, PERK, and XBP-1, suggesting that these ER stress molecules can act mainly as pathological factors in AD.
Lines 216-217: Therefore, the target molecules of ER stress for ALS are GRP78/BiP, CHOP, and XBP-1, suggesting that these ER stress molecules can act mainly as pathological factors in ALS.
Lines 251-253: Therefore, the target molecules of ER stress for TSEs are GRP58, GRP78/BiP, and PERK, suggesting that these ER stress molecules can act mainly as pathological factors in TSEs.
Lines 270-272: Therefore, the target molecules of ER stress for polyglutamine diseases are ASK1, GRP78/BiP, CHOP, and ERAD, suggesting that these ER stress molecules can act mainly as pathological factors in polyglutamine diseases.
Lines 305-307: Therefore, the target molecules of ER stress for LSD are GRP78/BiP, CHOP, PERK, IRE1, ATF6, and ATF4, suggesting that these ER stress molecules can act mainly as pathological factors in LSD.
Lines 321-323: Therefore, the target molecules of ER stress for acute neurodegeneration are GRP94, CHOP, and ATF4, suggesting that these ER stress molecules can act mainly as pathological factors in acute neurodegeneration.
Lines 325-332: Melatonin is a potent antioxidant and free radical scavenger [111, 112] and pro-tects inflammation, apoptosis, or autophagy in physiological and pathophysiological conditions [113-116]. In addition, melatonin modulates ER stress and UPR dysfunction in cancers, liver diseases, and other pathologies [115-117]. In particular, review articles introduce that melatonin can suppress various diseases related to ER stress: chronic intestinal inflammation and colon cancer [118], breast cancer [119], osteoarthritis [120], hepatocellular carcinoma [211], delaying ovarian aging [122], acute myocardial infarction [123], and diabetic cardiomyopathy [124].
Lines 701-733:
- Bonnefont-Rousselot, D.; Collin, F.; Jore, D.; Gardès-Albert M. Reaction mechanism of melatonin oxidation by reactive oxygen species in vitro. Pineal Res. 2011, 50, 328-335. doi: 10.1111/j.1600-079X.2010.00847.x.
- Galano, A.; Tan, D.X.; Reiter, R.J. Melatonin as a natural ally against oxidative stress: a physicochemical examination. Pineal Res. 2011, 51, 1-16. doi: 10.1111/j.1600-079X.2011.00916.x.
- Kim, S.H.; Lee, S.M. Cytoprotective effects of melatonin against necrosis and apoptosis induced by ischemia/reperfusion injury in rat liver. Pineal Res. 2008, 44, 165-171. doi: 10.1111/j.1600-079X.2007.00504.x.
- Muñoz-Casares, F.C.; Padillo, F.J.; Briceño, J.; Collado, J.A.; Muñoz-Castañeda, J.R.; Ortega, R.; Cruz, A.; Túnez, I.; Montilla, P.; Pera, C.; Muntané, J. Melatonin reduces apoptosis and necrosis induced by ischemia/reperfusion injury of the pancreas. Pineal Res. 2006, 40, 195-203. doi: 10.1111/j.1600-079X.2005.00291.x.
- Tuñón, M.J.; San-Miguel, B.; Crespo, I.; Laliena, A.; Vallejo, D.; Álvarez, M.; Prieto, J.; González-Gallego, J. Melatonin treatment reduces endoplasmic reticulum stress and modulates the unfolded protein response in rabbits with lethal fulminant hepatitis of viral origin. J Pineal Res. 2013, 55, 221-228. doi: 10.1111/jpi.12063.
- San-Miguel, B.; Crespo, I.; Vallejo, D.; Álvarez, M.; Prieto, J.; González-Gallego, J.; Tuñón, M.J. Melatonin modulates the autophagic response in acute liver failure induced by the rabbit hemorrhagic disease virus. Pineal Res. 2014, 56, 313-321. doi: 10.1111/jpi.12124.
- Fernández, A.; Ordóñez, R.; Reiter, R.J.; González-Gallego, J.; Mauriz, J.L. Melatonin and endoplasmic reticulum stress: relation to autophagy and apoptosis. Pineal Res. 2015, 59, 292-307. doi: 10.1111/jpi.12264.
- Motilva, V.; García-Mauriño, S.; Talero, E.; Illanes, M. New paradigms in chronic intestinal inflammation and colon cancer: role of melatonin. Pineal Res. 2011, 51, 44-60. doi: 10.1111/j.1600-079X.2011.00915.x.
- Naziroğlu, M.; Tokat, S.; Demirci, S. Role of melatonin on electromagnetic radiation-induced oxidative stress and Ca2+ signaling molecular pathways in breast cancer. Recept. Signal. Transduct Res. 2012, 32, 290-297. doi: 10.3109/10799893.2012.737002.
- Hosseinzadeh, A.; Kamrava, S.K.; Joghataei, M.T.; Darabi, R.; Shakeri-Zadeh, A.; Shahriari, M.; Reiter, R.J.; Ghaznavi, H.; Mehrzadi, S. Apoptosis signaling pathways in osteoarthritis and possible protective role of melatonin. Pineal Res. 2016, 61, 411-425. doi: 10.1111/jpi.12362.
- Mortezaee, K. Human hepatocellular carcinoma: Protection by melatonin. Cell. Physiol. 2018, 233, 6486-6508. doi: 10.1002/jcp.26586.
- Yang, Y.; Cheung, H.H.; Zhang, C.; Wu, J.; Chan, W.Y. Melatonin as Potential Targets for Delaying Ovarian Aging. Drug Targets 2019, 20, 16-28. doi: 10.2174/1389450119666180828144843.
- Fu, Z.; Jiao, Y.; Wang, J.; Zhang, Y.; Shen, M.; Reiter, R.J.; Xi, Q.; Chen, Y. Cardioprotective Role of Melatonin in Acute Myocardial Infarction. Physiol. 2020, 11, 366. doi: 10.3389/fphys.2020.00366.
- Huang, K.; Luo, X.; Zhong, Y.; Deng, L.; Feng, J. New insights into the role of melatonin in diabetic cardiomyopathy. Res. Perspect. 2022, 10, e00904. doi: 10.1002/prp2.904.Melatonin
**All of the edited sections and references were changed with the blue words.
***The Certificate of Editing is attached.
I hope that the revised manuscript is now acceptable for publication in the IJMS. We are looking forward to receiving your answer soon.
Sincerely,
Yeong-Min Yoo Ph.D
College of Life Science,
Gangneung-Wonju National University,
Gangneung, Gangwon-do 25457, Republic of Korea
Email: yyeongm@hanmail.net

Round 2
Reviewer 3 Report
The manuscript can be accepted for publication as per editor's final discretion. Thank you.